# Deep Eutectic Liquids as a Topical Vehicle for Tadalafil: Characterisation and Potential Wound Healing and Antimicrobial Activity

**DOI:** 10.3390/molecules28052402

**Published:** 2023-03-06

**Authors:** Bayan Alkhawaja, Faisal Al-Akayleh, Ashraf Al-Khateeb, Jehad Nasereddin, Bayan Y. Ghanim, Albert Bolhuis, Nisrein Jaber, Mayyas Al-Remawi, Nidal A. Qinna

**Affiliations:** 1Department of Pharmaceutical Medicinal Chemistry and Pharmacognosy, Faculty of Pharmacy and Medical Sciences, University of Petra, Amman 11196, Jordan; 2Department of Pharmaceutics and Pharmaceutical Technology, Faculty of Pharmacy and Medical Sciences, University of Petra, Amman 11196, Jordan; 3Depratment of Pharmaceutical Sciences, Faculty of Pharmacy, Zarqa University, Zarqa 13110, Jordan; 4University of Petra Pharmaceutical Center, Faculty of Pharmacy and Medical Sciences, University of Petra, Amman 11196, Jordan; 5Department of Life Sciences, Centre for Therapeutic Innovation, University of Bath, Bath BA2 7AY, UK; 6Faculty of Pharmacy, Al-Zaytoonah University of Jordan, Amman 11733, Jordan

**Keywords:** deep eutectic solvents (DESs), ionic liquids (ILs), tadalafil, wound healing, lidocaine, topical application, drug delivery system

## Abstract

Deep eutectic solvents (DESs) and ionic liquids (ILs) offer novel opportunities for several pharmaceutical applications. Their tunable properties offer control over their design and applications. Choline chloride (CC)-based DESs (referred to as Type III eutectics) offer superior advantages for various pharmaceutical and therapeutic applications. Here, CC-based DESs of tadalafil (TDF), a selective phosphodiesterase type 5 (PDE-5) enzyme inhibitor, were designed for implementation in wound healing. The adopted approach provides formulations for the topical application of TDF, hence avoiding systemic exposure. To this end, the DESs were chosen based on their suitability for topical application. Then, DES formulations of TDF were prepared, yielding a tremendous increase in the equilibrium solubility of TDF. Lidocaine (LDC) was included in the formulation with TDF to provide a local anaesthetic effect, forming F01. The addition of propylene glycol (PG) to the formulation was attempted to reduce the viscosity, forming F02. The formulations were fully characterised using NMR, FTIR and DCS techniques. According to the obtained characterisation results, the drugs were soluble in the DES with no detectable degradation. Our results demonstrated the utility of F01 in wound healing in vivo using cut wound and burn wound models. Significant retraction of the cut wound area was observed within three weeks of the application of F01 when compared with DES. Furthermore, the utilisation of F01 resulted in less scarring of the burn wounds than any other group including the positive control, thus rendering it a candidate formula for burn dressing formulations. We demonstrated that the slower healing process associated with F01 resulted in less scarring potential. Lastly, the antimicrobial activity of the DES formulations was demonstrated against a panel of fungi and bacterial strains, thus providing a unique wound healing process via simultaneous prevention of wound infection. In conclusion, this work presents the design and application of a topical vehicle for TDF with novel biomedical applications.

## 1. Introduction

Ionic liquids (ILs) are usually defined as a neoteric class of molten salts with a liquid state at temperatures below 100 °C, with some of them being liquid at room temperature. They comprise discrete inorganic or organic anions and organic cations with favourable solvation properties, low vapour pressure, chemical adaptability, and thermal stability [1,2]. Due to their combination of organic character with ionic nature, ILs have been adopted as alternatives to organic solvents with ubiquitous applications [3,4].

Deep eutectic solvents (DESs) have received huge amounts of attention as green substitutes to ILs, sharing most of the unique properties of ILs. Some reports consider DESs to be a new class of ILs [5]. However, DESs have superior desirable properties compared to ILs, such as lower preparation cost, facile preparation techniques, non-toxicity and biodegradability [6]. DESs are usually formed by direct mixing of Lewis or Brønsted acids and bases in specific ratios. The formed systems have noticeably lower melting points compared to their individual components [7,8]. DESs formed between quaternary ammonium salts and carboxylic acids are amongst the most investigated systems [9].

As one of the most versatile types of DESs, choline chloride salt-based DESs (referred to as Type III eutectics) have been widely utilised in chemical applications, such as catalysis, extraction from biomass, and synthesis reactions [6,10,11,12]. With the unique advantages of choline-based DESs, notably an interest in their tunable biological applications, they have been extensively evaluated in a myriad of pharmaceutical applications [13,14,15,16]. As an alternative to organic solvents, DESs have provided green alternatives with various medical implications, such as providing vehicles for transdermal drug delivery and wound healing applications [17,18,19].

Non-healing or chronic wounds impose an enormous burden on and expenditure within global health sectors [20,21]. Wound healing refers to a dynamic biological process comprising four main physiological stages, namely, haemostasis, inflammation, proliferation, and tissue remodeling. Wound healing phases should be sequenced and stewarded timely [22]. Various local or systemic factors could interfere with one or more stages of wound healing, which ultimately could impair the healing process or lead to the development of chronic wounds [22,23]. One of the main factors that impairs the normal wound healing process is hypoxia, which, although initially essential for healing, is detrimental for wound healing if prolonged [24].

The merits of using selective phosphodiesterase type 5 (PDE-5) enzyme inhibitors in the wound healing process have been widely investigated. PDE-5 is an enzyme involved in the degradation of cyclic guanosine monophosphate (cGMP), and its inhibitors have a vasodilation effect. In addition, they also prolong the effect of nitric oxide (NO) both cellularly and endovascularly [25,26,27]. NO is an important signaling molecule that could facilitate the wound healing process via several mechanisms, including promoting cellular proliferation, tissue remodeling and angiogenesis [28]. Amongst the selective PDE-5 inhibitors, tadalafil (TDF) has shown an intriguing PDE selectivity [27,29].

Previously, Alwattar et al. adopted spray-dried TDF in wound healing [27]. Moreover, orally delivered TDF was tested in a porcine burn model and shown to result in faster reepithelialisation and reduced scarring [30]. In addition, a positive impact of oral PDE-inhibitors in skin flap healing viability was demonstrated in a rat model [31]. However, oral administration of TDF causes a range of adverse and undesirable side effects, such as headache, myalgia, and flushing [32]. Therefore, the topical application of TDF could be a better solution for wound management.

It is well established that improving the delivery mechanisms and pharmacokinetic characteristics of existing drugs is financially preferable to the launching of a new one [33,34]. Hence, we set out to investigate choline chloride-based DES as a carrier vehicle for TDF and to repurpose TDF for wound management. Moreover, to obtain an optimal healing process, lidocaine (LDC) was adopted in the wound management formula. LDC is a local anaesthetic drug; hence, it was incorporated to manage the pain associated with wounds [35,36].



The advantages of the use of topical PDE-5 inhibitors combined with the beneficial properties of DES-based drug delivery represent the rationale behind the wound healing method investigated in this work. Herein, this work demonstrated the utility of DESs of TDF and LDC as a strategy for topical application as well as for the management of wounds, while possessing good antimicrobial activity.

## 2. Results and Discussion

### 2.1. Preparation of DESs and Drug-Loaded DES Formulations

Initially, we set out to validate the suitability of the DES formulations for topical application. In general, topically applied products entail adequate rheological behaviour, spreadability, and appropriate skin tolerability [37]. Rheological testing of topically applied products is essential for product evaluation. Generally, these experiments are performed primarily to ascertain suitability for product manufacturing purposes, uniformity, extrudability, stability and topical applicability [38]. Although decreasing the viscosity would enhance the applicability and penetration of the topical product, choosing the optimum viscosity usually varies depending on the purpose of the treatment and the final topical product.

To this end, different compositions of blank DES were prepared using various molar equivalents of malonic acid (MA) to CC (1:1, 1:2, and 2:1), with or without propylene glycol (PG) (Table 1).

The rheological study results are presented in Figure 1 and compared to a commercially available topical cream for wounds, ialuset Plus. In order of increasing viscosity the formulations ranked B03 ˃ B01 ˃ B02. The relatively high viscosity of B03 hinders its topical application as a drug carrier system; hence, it was not optimal for the purpose of our study. On the other hand, B01 showed lower viscosity behaviour than B02 at room temperature (25 °C) and therefore could be a candidate blank formula (Figure 1)

To further enhance the fluidity of the DESs, the co-solvent PG was incorporated at different ratios. PG has been widely utilised in topical applications, notably to enhance drug permeation through the skin [39]. The rheological behaviour of DESs with PG was compared with ialuset Plus. The viscosity of B04 was closer to that of the commercial product at 25 °C at a shear rate of 50 s^−1^ (360 ± 7.1 MPa and 380 ± 5.1 MPa for B04 and ialuset Plus, respectively) (Figure 1).

Next, the contact angle was measured for DESs with PG, with a lower contact angle indicating a less hydrophobic character. The incorporation of PG resulted in a significant reduction in the DES contact angles on the hydrophobic glass surface, with B04 and B05 formulations being the lowest (Table 1). Owing to the suitable viscosity behaviour for topical application, B04 formulation showed close spreadability readings when compared with the commercial product (6.0 ± 0.12 and 6.3 ± 0.11 cm, respectively) (Table 1). Based on the aforementioned results with respect to the commercial product, B01 and B04 could be candidate formulations to be adopted for topical application purposes.

Having evaluated the optimum DESs for topical application, we then studied the equilibrium solubility of TDF in B01. Accordingly, the equilibrium solubility of TDF in B01 (4.3 mg/mL) was found to be 1133-fold higher when compared with the aqueous solubility of TDF at room temperature (4.3 mg/mL and 0.003 mg/mL, respectively) [40]. Indeed, TDF is practically insoluble in water [40,41] and, hence, a significant improvement in the saturated solubility in DES could be a significant advantage for enhancing the bioavailability of TDF. The latter is, however, beyond the aim of this study.

LDC belongs to the class I drugs, which have high solubility and permeability according to the biopharmaceutical classification system [42]. LDC could have an effect on the solubility of TDF in DES, which was therefore evaluated using increasing molar rations of LDC:TDF. The equilibrium solubility of TDF in B01 was evaluated separately and likewise with increasing molar ratios of LDC. A notable improvement in the solubility of TDF in DES was observed upon addition of LDC, with an almost twofold increase in solubility when mixed at 3:1 molar ratio (Figure 2). Our results were in accordance with previous findings reported by Marei and co-workers, where LDC was shown to enhance the solubility of a group of nonsteroidal anti-inflammatory drugs [43]. Hence, in this work, LDC was added to the formulation as a local anaesthetic to ease the pain associated with wounds; in addition, we demonstrated that it enhanced the solubility of TDF in the blank DES. Hence, LDC exhibited double action by acting as a solubility enhancer and being pharmacologically active as a topical anaesthetic.

The prepared drug formulations within this work and their composition are illustrated in Table 2. Given that B01 and B04 gave optimal topical behaviour comparable with the commercial drug, two main formulations were prepared: formulation 1 (F01) of TDF and LDC was prepared with B01 as a vehicle, while F02 was prepared using B04 as a vehicle. Lastly, F03 was prepared using TDF without LDC to assess the impact of LDC on the wound healing process.

### 2.2. Characterisation of DES Formulations

Detailed characterisation of the prepared formulations containing TDF and LDC was performed using a range of analytical techniques, including NMR, ATR-FTIR, and DSC.

#### 2.2.1. Nuclear Magnetic Resonance (NMR)

Analyses were conducted to investigate the structural characteristics of the blank and drug-loaded formulations (Figure 3). However, as the final concentrations of both TDF and LDC were considerably lower in the tested samples, well-pronounced peaks corresponding to the DES were observed in the ^1^H NMR spectrum. Therefore, detailed structural assignments of the chemical shifts associated with the structure of the studied drugs were unattainable even after the solvent suppression method was employed. Nevertheless, with the expansion around the aromatic region (6–8 ppm), peaks corresponding to both TDF and LDC chemical structures were detected in the expected ratios, confirming the presence of both drugs without any degradation after their incorporation into the DES (Figure 3C). Moreover, ^1^H NMR analysis of the individual drugs was conducted to further ascertain these observations, and the results were in accordance with the DES formulation loaded with the drugs (Appendix A).

#### 2.2.2. Attenuated Total Reflectance—Fourier Transform Infrared Spectroscopy (ATR-FTIR)

Figure 4A shows the FTIR spectra of MA, CC, the blank DES formulation, TDF, LDC, and formulation 1. The most notable peaks in the FTIR spectrum of CC were the peaks at 3220 cm^−1^ and 1482 cm^−1^, corresponding to OH stretching and bending peaks of CC respectively, as well as the peak at 1348 cm^−1^, likely corresponding to C–N bending [44]. In the case of MA, the peaks at 3220 cm^−1^ and 3287 cm^−1^ corresponded to the OH stretching of the two hydroxyl groups, and the peaks observed at 1720 cm^−1^ and 1688 cm^−1^ corresponded to the C=O stretching of the two carbonyl groups present. Other notable peaks in the MA spectrum were the peaks at 1390 cm^−1^ and 1418 cm^−1^, which correspond to OH bending. In the B01 spectrum, peaks of note were the broad peak centered around 2936 cm^−1^, with notable shoulder peaks at 3300 cm^−1^ and 3031 cm^−1^, which likely corresponds to the shifted OH stretching peaks of CC and MA. Furthermore, a notable peak was seen 1at 1718 cm^−1^, likely corresponding to the shifted C=O stretching of MA. The OH bending peaks of MA were observed at 1379 cm^−1^ and 1417 cm^−1^, along with the peak at 1478 cm^−1^, likely corresponding to the shifted OH bending peak of CC. The weak C–N stretching peak of CC was not visible in the DES spectrum. This suggests that the mechanism of formation of the DES is likely due to hydrogen bonding mediated by the C=O groups of MA with the OH group of CC. Characteristic peaks in the LDC spectrum were the NH stretching peak seen at 3454 cm^−1^, the C=O stretching peak at 1672 cm^−1^, and the NH bending peak at 1656 cm^−1^. Characteristic peaks in the TDF spectrum were the peaks seen at 3326 cm^−1^, 1676 cm^−1^, 1646 cm^−1^, and 1628 cm^−1^, which likely correspond to NH stretching, stretching of the two C=O groups, and NH bending, respectively. The FTIR spectrum of F01 (containing the DES in which TDF and LDC were dissolved) was largely similar to the spectrum of the placebo DES, with the previously reported peaks at 3300 cm^−1^, 3031 cm^−1^, and 1718 cm^−1^. No peaks which are characteristic of either drug were visible in the spectra, most likely due to the concentration of both drugs in the formulation being below the limits of detection by ATR-FTIR.

Figure 4B shows the spectra of F01, PG, and F02. The characteristic peaks of PG were seen at 3311 cm^−1^, 1459 cm^−1^, and 1412 cm^−1^, corresponding to OH stretching and OH bending, respectively. In the spectrum of the viscosity-adjusted formulation (F02), the aforementioned C=O stretching peak of the B01 formulation was seen to shift to 1721 cm^−1^. The OH stretching peak of PG was also visible, albeit shifted to 3326 cm^−1^, which is suggestive of hydrogen bonding between the DES formulation and PG.

#### 2.2.3. Differential Scanning Calorimetry (DSC)

Figure 5A shows thermograms of the raw materials (CC, MA, TDF, and LDC) during the first heating scan. The melting point (T_m_) of CC was seen at 253 °C. The T_m_ of MA was seen at 136 °C. The T_m_ of LDC was seen at 80 °C, and the melting point of TDF was found to be around 303 °C.

In the cooling cycle (Figure 5B), no recrystallisation was observed for MA, TDF, and LDC. Furthermore, glass transitions (T_g_) were observed for both LDC (at −15 °C) and TDF (at 129 °C). An exothermic event indicative of recrystallisation was seen at 53 °C in the thermogram of MA (enthalpy: 1.81 J/g, corresponding to ≈0.8% recrystallisation of the 214.47 J/g observed in the first heating cycle).

In the second heating cycle (Figure 5C), an endothermic peak indicative of melting was seen in the thermogram of MA at 89 °C (enthalpy: 1.59 J/g, corresponding to ≈0.7%). The amount recrystallised that was observed in the cooling cycle (Figure 5B) corresponds to the amount recovered in the second heating cycle (Figure 5C), suggesting that the observed event is indeed the recrystallisation and melting of MA. The depression of the T_m_ of MA from 136 °C to 89 °C is likely due to the excessive amorphisation of MA in DSC (>99%) causing depression in MA melting. The aforementioned T_g_ of TDF was clearly visible in the second cooling cycle.

Figure 5D shows DSC thermograms of B01, F01, and F02. No endothermic events corresponding to the melting of either of the DES components MA and CC were seen, nor any endothermic events indicative of the presence of either drug in the crystalline form.

The results obtained from DSC, coupled with FTIR data, suggest that at the aforementioned ratio, the MA:CC interaction results in the formation of a homogenous, single-phase liquid with no signs of recrystallisation of either material. Furthermore, the absence of any endothermic events is indicative of complete solubilisation of both TDF and LDC in the DES formulation.

Furthermore, due to the contact-dependent nature of ATR-FTIR, any crystalline content within the DES formulations would have been significantly more prominent than what was observed in Figure 4B. This suggests that, within the limits of detection of DSC and FTIR, no crystalline content is present within the formulation, with all constituents (MA, CC, LDC, and TDF) appearing to have formed a homogenous, single-phase solution. Moreover, PG does not appear to have any negative influence on the solubilisation potential of the DES formulation, as no changes in thermal behaviour were seen in DSC upon the addition of PG.

### 2.3. Stability Evaluation of the TDF Formulations

The stability of the F01 and F02 formulations was determined after incubation at ambient temperature for three months. No precipitate was noticed, nor any change in the physical appearance of the formulation under ambient storage conditions. More importantly, the quantity of TDF was assayed with HPLC in F01 and F02, and was found to be 84% and 91%, respectively, of the amount present at the start of the experiment. These results reflect that TDF and LDC exhibited excellent stability in both formulations under ambient storage conditions.

### 2.4. Effect of DES Formulations on Cut Wound Healing Process

TDF is a PDE-5-inhibitor primarily employed for erectile dysfunction. By inhibition of PDE and elevating the level of cGMP, the effects of NO, including relaxation of smooth muscle cells and enhancing vasodilation, will dominate. NO is a central player in the wound-healing process through modulating vascular homeostasis and inflammation, and has antimicrobial effects. Previous studies demonstrated the positive impact of oral TDF in the burn healing process [21,45]. We proposed that the topical application of TDF could improve the wound healing process without the need for systemic exposure to the drug, whereas LDC could help in relieving the pain associated with wounds without any negative impact on the healing process, as previously demonstrated [46]. This was tested using a rat wound-healing model. In the assessment of cut wound healing, F01 showed retraction of the wound area at week 3, unlike B01, which showed a slight decrease in the wound area at week 4. F01 formed scars at week 4, while B01 formed scars at week 5. Scars disappeared at week 6 in both treated groups. Pus formation was observed in both treated groups until week 2 (Figure 6 and Figure 7).

The group treated with B04 (DES with PG) showed significant retraction in wound size not earlier than the third week of dosing, which left a scar at the following week but disappeared after five weeks of treatment. In regard to animals treated with F02, mild wound retraction was observed at the third week of dosing. However, scar formation persisted during the following two weeks and disappeared after six weeks of treatment (Figure 6 and Figure 7).

Treatment with F03 (TDF alone) showed some lag time in healing until week 4, when a significant retraction in wound size was noted. Marks of wound were absent at week 5, when no scar formation was noted (Figure 6 and Figure 7).

It could be concluded that the prepared formulations with TDF were not very effective for cut wound healing when compared with the blank B04. The most consistent recovery of cut wounds was observed in group B04, unlike other groups, which showed mild effects in retracting the wound area within the first three weeks of treatment. The positive control (ialuset Plus) showed a fast recovery rate in comparison to all treatments. The positive control showed a consistent and clean (no pus) skin healing process, and also showed mild formation of scars in the third week which disappeared in the fourth week. The sham group (no treatment) healed well within the first three weeks.

### 2.5. Effect of DES Formulations on Burn Wound Healing Process

Regarding burn wound healing, the use of all treatments on burn wounds showed significant retraction and healing at the third week, except for F03, which showed no significant change on burn wounds until the fourth week. At week four, small wounds were evident, which formed scars at the fifth week. Scars persisted in all groups until the sixth week, except for F01, ialuset Plus and F03, as they were found to be negligible in those groups (Figure 8). The morphological changes within the skin layers are shown in Figure 9; wounds treated with F01 showed formation of serohematic scabs that did not persist and peeled off indicating healthy, repaired skin underneath, as confirmed by histopathological examinations at day 28 of the study.

Collectively, F01 showed less scarring of burn wounds than any other group including the positive control, thus being a candidate formula for burn dressing formulations (Figure 8 and Figure 9). A rapid wound healing process of the skin has been reported to cause formation of fibrotic scar tissue [47]. Therefore, the slow healing process of F01 was found to be the preferred treatment for burn healing.

When compared with F02, F01 exhibited slower healing results, which could be related to the presence of PG in F02. Previously, PG formulation was found to be ineffective for the skin wound healing process [48]; however, it has been utilised in formulations, notably to enhance drug permeation through the skin from topical preparations [39]. In F02, PG was incorporated in a 1:1 ratio to enhance the spreadability of the treatment, which was found to be effective for rapid retraction and healing of burn wounds at the third week. This could be related to the enhanced penetration of the TDF through the intact skin. Nevertheless, in our hands, we found that our treatments (F01 and F02) showed slower healing processes when compared with the commercial product, and F01 exhibited less scarring than any other group, which is more preferred in burn healing.

### 2.6. Antimicrobial Activity Testing (In Vitro)

Generally, most DESs have antibacterial activity against a wide range of Gram-positive and Gram-negative bacteria. According to Zakrewsky and co-workers, DES of CC and MA exhibited strong activity against bacterial biofilms of both *Pseudomonas aeruginosa* and *Salmonella enterica* serovar Typhimurium [19].

Skin wounds are usually complicated by the presence of microbes, ranging from contamination and colonisation to invasive infection [23]. Therefore, we set out to evaluate the antimicrobial activity of the developed formulations. The antimicrobial activity of DES and formulation 1 were tested against a panel of bacterial and fungal strains, and their MIC values are reported in Table 3. This showed that good antibacterial activity was observed with B01, which was not significantly altered by the presence of TDF and LDC. However, B01 was not active against the yeast *C. albicans*.

## 3. Materials and Methods

### 3.1. Materials

Tadalafil (TDF) and lidocaine HCl (LDC) were kindly gifted by the Jordanian Pharmaceutical Company (JPM) Amman, Jordan. Choline chloride and malonic acid were purchased from Tokyo Chemical Industry Co., Ltd. (TCI), Tokyo, Japan.

### 3.2. HPLC Analysis

HPLC (Shimadzu LC20AT HPLC system, Shimadzu LTD, Japan) was employed for the determination of TDF concentration in the DES. The analysis method adopted for TDF was developed and validated in an earlier report [49]. Briefly, the mobile phase, consisting of acetonitrile and phosphate buffer (60%:40%) with pH 7 adjusted using phosphoric acid, was circulated through a C-18 column (150 × 4.6 mm) packed with a particle size of 5 μm. The flow rate was kept at 0.8 mL/min and the wavelength of detection was 262 nm. The standard curves of TDF and LDC were found to be linear (Appendix A).

### 3.3. Choline Chloride–Malonic Acid (Blank Formulations) DES Preparation

Different compositions of blank DES were prepared using various molar equivalents of malonic acid (MA) to choline chloride (CC) (1:1, 1:2, and 2:1), with or without propylene glycol (PG) (Table 1). The DES mixtures were prepared by mixing CC and MA; the mixtures were left for 24 h with continuous stirring at room temperature until a clear and homogenous liquid was obtained. This was then left to equilibrate in a shaking water bath for 12 h at 190 rpm and 25 °C. Lastly, the prepared DESs were mixed with propylene glycol (PG) at different ratios, as shown in Table 1.

### 3.4. Characterisation of Blank DESs

#### 3.4.1. Rheology Study

The viscosity of prepared DESs were studied using a rheometer (Physica MCR 302, Anton Paar, Austria) with different geometries (concentric cylinder, cone-and-plate, and parallel-plate), coupled with a Cp 50 double gap concentric cylinder measurement system. Initially, the instrument was calibrated, and each DES (5–10 mL) was loaded between the concentric cylinders. The measurement conditions were shear rate (0.1–100), temperature set (−10–32 °C), cone angle 1 and zero-gap 0.1.

#### 3.4.2. Contact Angle Measurements

The contact angles of a drop of DES were measured on a polyethylene plastic surface using a contact angle goniometer (OCA 15 EC, Data Physics instruments GmbH, Filderstadt, Germany) and analysed using SCA20 Software (, Dataphysics, Germany.) for optical contact angle (OCA) and portable contact angle meter (PCA) (PCA-1, Kyowa Interface Science, Niiza, Japan). For each measurement, a 500 µg Hamilton syringe was filled with the sample and anchored on the device. The dosing volume of each drop was 4 µg with a dosing rate of 1 µg/s. High-resolution images of each drop were captured using a fixed camera (Sony, Tokyo, Japan).

#### 3.4.3. Spreadability

The spreadability of the DESs was investigated by placing 1 g of the DES preparation in the center of a 20 × 20 cm glass plate. It was then covered with another slide and left for one minute. The diameter of the spread area (cm) was measured. The results are presented as the mean along with the standard deviation from three independent experiments.

### 3.5. Determination of TDF Solubility in DESs (Shake-Flask Technique)

For the determination of the equilibrium solubility of the TDF in a blank formulation, an excess amount of TDF was added to the DES. Then, the system was kept shaking (190 RPM) for 24 h in a thermostatic water bath at 25 °C. The excess (undissolved) TDF was separated by centrifugation (Stuart SCF1 Mini Centrifuge Spinner, Bosco M-24A centrifuge, Hamburg, Germany) at 14,000 RPM for 5 min. After suitable dilution, the drug concentration was determined using HPLC.

### 3.6. Enhancing TDF Solubility Using LDC

A fixed amount TDF was dissolved in DES with increasing molar ratios of LDC (0.5, 1, 2 and 3). The equilibrium solubility of TDF was determined as previously described using the HPLC method.

### 3.7. Stability Study of TDF Formulations

The stability study of the TDF formulations was determined after incubation at ambient temperature for three months. Physical appearance, colour change, and precipitation were evaluated. Chemical stability was determined using HPLC.

### 3.8. Nuclear Magnetic Resonance (NMR)

NMR datasets were collected on a 500-MHz Bruker instrument (Bruker Avance III, BRUKER)using DMSO-d6. NMR spectra for CC-MA DES and formulation containing the drugs are shown in Figure 3A–C.

NMR assignments for B01: ^1^H NMR (500 MHz, DMSO) δ 12.69 (s, 2H), 5.58 (s, 1H), 3.87–3.78 (m, 2H), 3.46–3.39 (m, 2H), 3.26 (s, 2H), 3.16 (s, 1H). ^13^C NMR (126 MHz, DMSO) δ 168.81, 67.40, 67.38, 67.36, 55.49, 53.60, 53.57, 53.54, 42.47.

### 3.9. Attenuated Total Reflectance—Fourier Transform Infrared Spectroscopy (ATR-FTIR)

ATR-FTIR spectra for the blank and final formulations were acquired using a Perkin Elmer UATR Spectrum (PerkinElmer Inc., Waltham, MA, USA) within the range 4000–550 cm^−1^. Spectra were acquired in absorbance mode, with a resolution of 2 cm^−1^ and 32 scans per sample. The spectra were exported in Comma Separated Values (CSV) format and analysed using Spectrogryph version 1.2.15. No pre-processing (PerkinElmer Inc., Waltham, MA, USA) was applied to the spectra before analysis.

### 3.10. Differential Scanning Calorimetry (DSC)

Differential scanning calorimetry (DSC) thermograms were acquired using a TA Q25 Discovery instrument (TA Instruments, Newcastle, DE, USA). Scans were acquired using a heat–cool–reheat cycle at a heating/cooling rate of 10 °C/min. Choline chloride and tadalafil were scanned from 25–320 °C during the first heating cycle, equilibrated at 320 °C for one minute, and then cooled to −20 °C, followed by a second heating cycle up to 320 °C. Malonic acid was scanned at a temperature range of 25–250 °C during the first heating cycle, and was then equilibrated at 250 °C before cooling to −20 °C, followed by a second heating cycle up to 250 °C. Lidocaine was analysed at a temperature range of 25–100 °C, followed by cooling down to −20 °C, and a second heating cycle up to 100 °C. All formulations were analysed at a temperature range of 25–320 °C for the first heating cycle, 320–−20 °C for the cooling cycle, and −20 °C–320 °C for the second heating cycle. Standard aluminium pans containing samples weighing 2–4 mg were used for all scans. Nitrogen purge gas was used for all runs, with a constant flow rate of 50 mL/min.

### 3.11. Wound Healing Model (In Vivo)

Forty-nine adult male Sprague Dawley rats with an average weight of 250 ± 20 g were housed at the Laboratory Animal Research Unit of the University of Petra Pharmaceutical Center (UPPC), Amman, Jordan. Rats were kept under controlled temperatures of 22 ± 2 °C, at a humidity 60 ± 5% and with a 12-h light/dark cycle. The study was conducted in accordance with the University of Petra Institutional Guidelines on Animal Use, which adopts the guidelines of the Federation of European Laboratory Animal Science Association (FELASA), ethical approval number (1A/1/2020).

Animals were grouped into seven groups (*n* = 7), each group receiving its corresponding treatments, in addition to a sham group that received no treatment and a positive control group treated with a commercially available topical cream, namely, ialuset Plus, a product of IBSA Farmaceutici, Lodi, Italy.

The back of each animal was shaved using an electronic clipper and skin was observed to confirm absence of any irritation or scars. The following day, animals were placed on a surgical board (Kent Scientific, 1116 Litchfield Street Torrington, CT 06790, CT, USA) and anaesthetised with 2.5% isoflurane (Hikma Pharmaceuticals, Amman, Jordan) using a general anaesthesia system (SomnoSuite, Kent Scientific Corporation, city, CT, USA). Upon anaesthesia, wound and burn sections were created on the backs of the animals. An excision wound was made using a skin punch (diameter size: 0.8 mm) while a burn wound was generated by thermal damage through direct application of heat by placing a hot 80 °C metal plate on the skin for 10 s, as described by Masson-Meyers [50].

A thin layer of each formulation was applied aseptically once daily for six consecutive weeks. Exactly 1 g quantity of tested formulation or reference product was applied and distributed over the wound using sterile gloved hands to mimic clinical conditions.

Animals were observed for six consecutive weeks for wound healing, scar formation and scar healing. Samples of the skin were collected weekly from one animal of each group and were fixed in buffered formalin for H&E staining and histopathology examination.

### 3.12. Statistical Analysis

To determine significance between groups, one-way ANOVA was used followed by post-hoc Tukey’s HSD test, unless stated otherwise, using IBM SPSS Statistics 25, IBM Corporation (New York, NY, USA). Values are expressed as mean ± standard error of the mean (SEM). Results were considered significant for *p*-values ≤ 0.05. An independent samples *t*-test was conducted to analyse significance between candidate groups, in comparison to sham and positive control groups individually.

### 3.13. Antimicrobial Activity Testing (In Vitro)

The strains used were *Escherichia coli* MC4100 [51], *Pseudomonas aeruginosa* PA01 [52], *Staphylococcus aureus* NCTC6571, *Enterococcus faecalis* ATCC19422, and *Candida albicans* SC5314 [53]. The minimal inhibitory concentrations (MICs) were evaluated with a microbroth dilution method [45] using Mueller Hinton broth (MHB) for bacteria, or MHB supplemented with 2% glucose for *C. albicans.* The MIC values, expressed as µL/mL, were defined as the lowest concentration where no growth was visible after 24 h of incubation at 37 °C.

## 4. Conclusions

The work yielded an effective formulation for wound healing, particularly burn wounds. CC-based DES was prepared and the selection criteria for the blank formulations suitable for topical application was compared with a commercial product for wound healing, ialuset Plus. TDF exhibited high equilibrium solubility in DES formulations, and the developed formulations were fully characterised to ensure the suitability of this vehicle for applying TDF topically. The facile preparation approach of the formulations along with the long-term stability are amongst the main pros of this work. The combination of LDC and TDF was found to be optimal for the devoted aims of this work, as the former could provide a local anaesthetic effect without impairment of the healing process. The attained in vivo results for F01 were comparable to the commercial product in the healing time of cuts as well as burn wounds, and resulted in less scarring in burn wounds than other groups, including the commercial product. Lastly, the antimicrobial activity of the formulations containing DESs could be advantageous to counteract possible minor microbial infections associated with wounds.

## Figures and Tables

**Figure 1 molecules-28-02402-f001:**
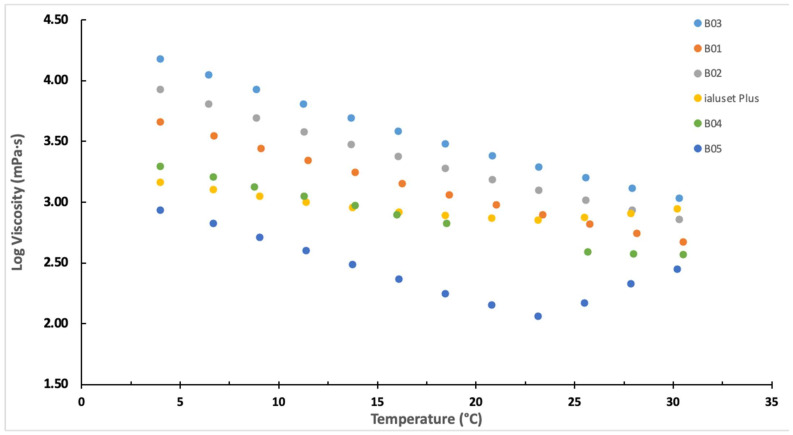
The rheological behaviour of the blank DESs compared with the commercial product (ialuset Plus) employed for wound healing, showing the viscosity behaviours of the developed formulations in comparison to ialuset Plus as the temperature increases up to 30 °C. B04 and B01 exhibited relatively equivalent viscosity at 25 °C.

**Figure 2 molecules-28-02402-f002:**
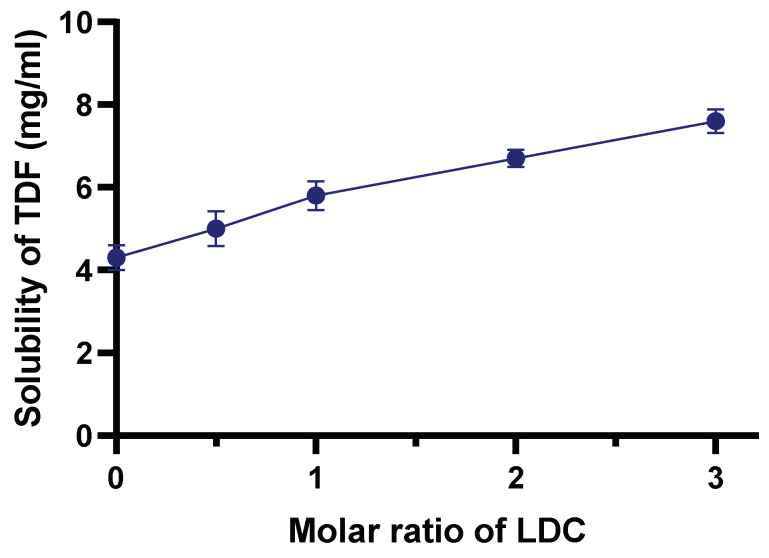
Solubility behaviour of TDF with increasing molar ratios of LDC. The results are represented as means along with standard deviation from three independent experiments.

**Figure 3 molecules-28-02402-f003:**
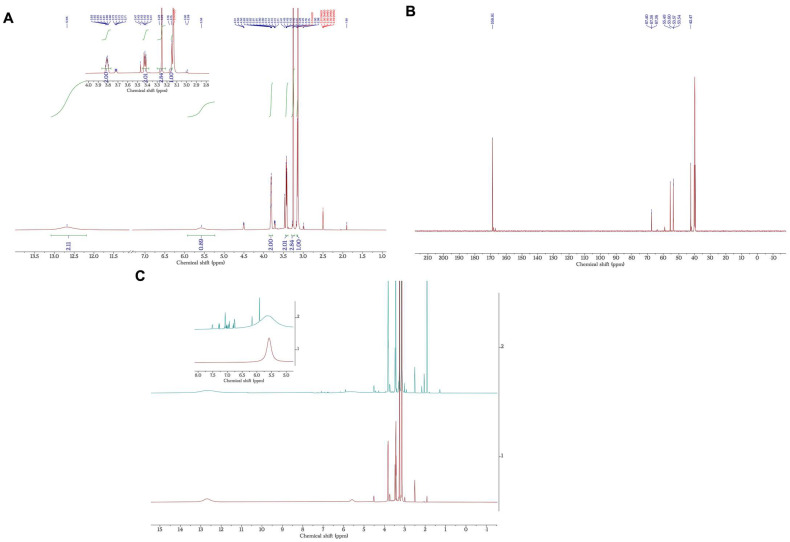
NMR spectra of B01 and F01, (**A**) ^1^H NMR of the B01 showing peaks corresponding to choline chloride and malonic acid in the expected 1:1 ratio, (**B**) ^13^C NMR of the B01, (**C**) Stacked ^1^H NMR spectra of B01 and F01, showing the inset for the aromatic region between 5 to 8 ppm.

**Figure 4 molecules-28-02402-f004:**
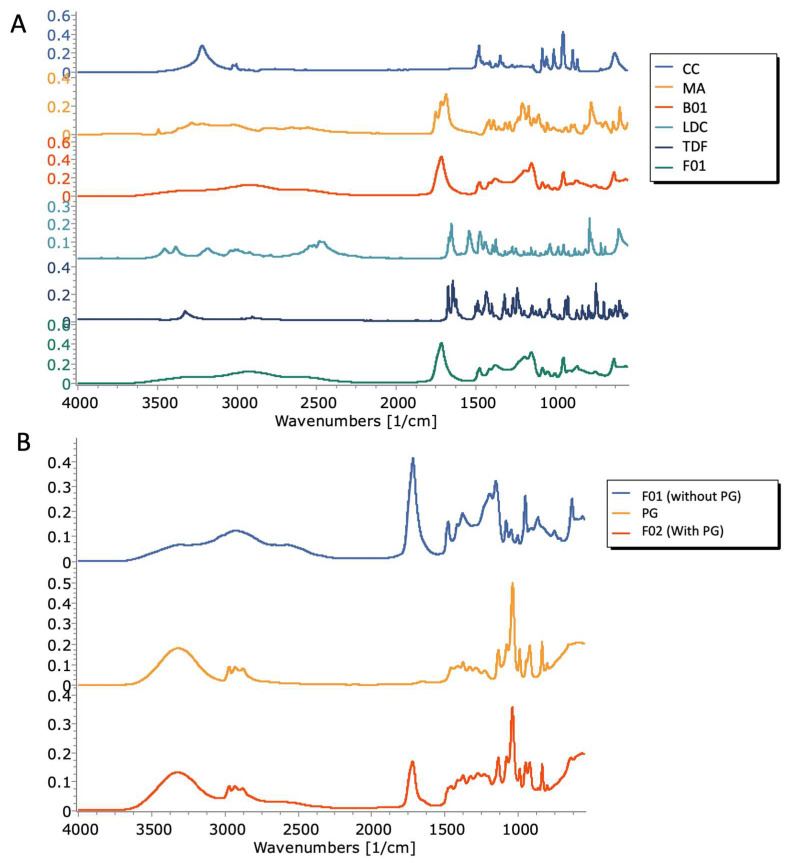
(**A**) FTIR spectra of MA, CC, B01, TDF, LDC, and F01; (**B**) FTIR spectra of F01, PG, and F02.

**Figure 5 molecules-28-02402-f005:**
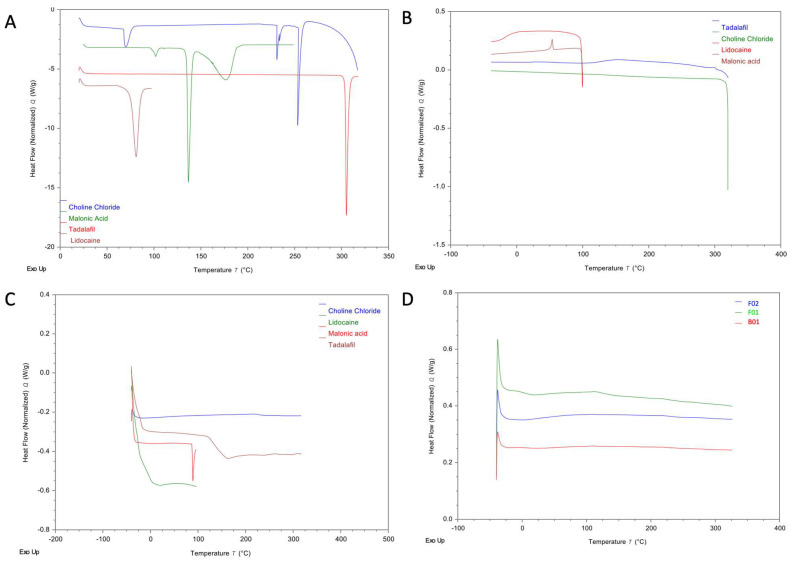
DSC thermograms of the raw materials: (**A**) 1st heating cycle, (**B**) cooling cycle, and (**C**) second heating cycle. (**D**) DSC thermograms of the placebo DES (B01), F01, and F02.

**Figure 6 molecules-28-02402-f006:**
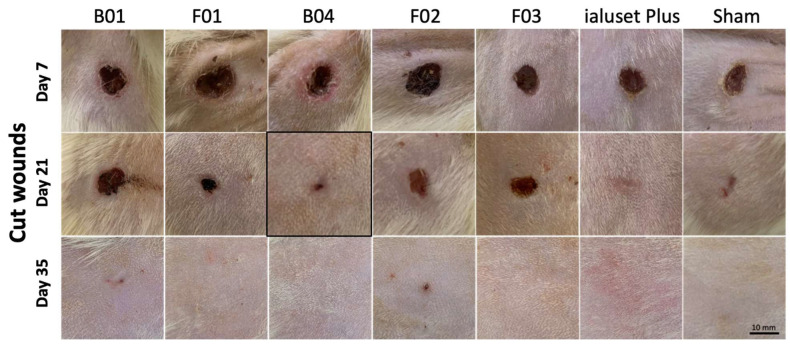
Pictures of cut wound healing on Sprague Dawley rats receiving DES formulations. Wounds of study groups observed at weeks 1 (day 7), 3 (day 21) and 5 (day 35). Unhealed wounds at week 1, with significant retraction of cut area in groups receiving positive controls F01 and B04. Mild retraction of cut wounds in animals receiving F02 and F03. No retraction in cut area of animals receiving B01 at week 3. At week 5, full healing without scarring in all groups except in animals receiving F02 and B01. Photographs in boxes are those most clinically healed at time of observation in comparison to the positive control.

**Figure 7 molecules-28-02402-f007:**
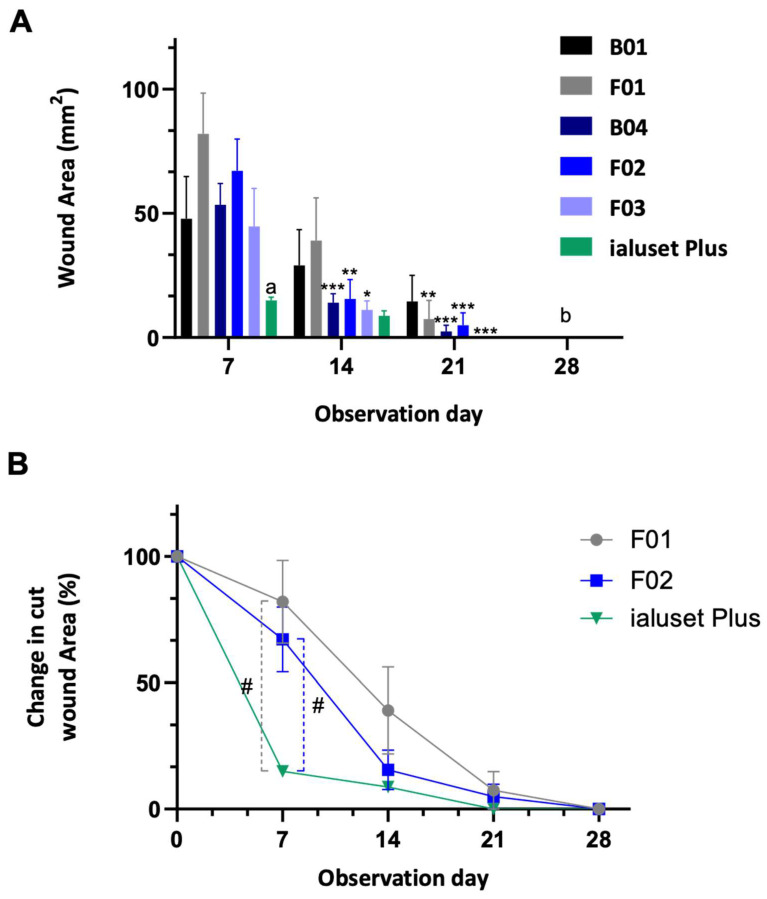
Retraction of cut wound area; 100% represents the baseline of the cut area at the start of the dosing period. Data represented as mean ± SEM (*n* = 7). *, **, ***: Statistical difference in comparison to day 7 with *p*-values < 0.05, < 0.01, < 0.001, respectively. (**A**) Statistical difference (*p*-value < 0.05) between wound mass of ialuset Plus at each observation day and all formulations at day 7. (**B**) all wound masses were significantly retracted in comparison to previous days, *p*-value < 0.01. #: significant.

**Figure 8 molecules-28-02402-f008:**
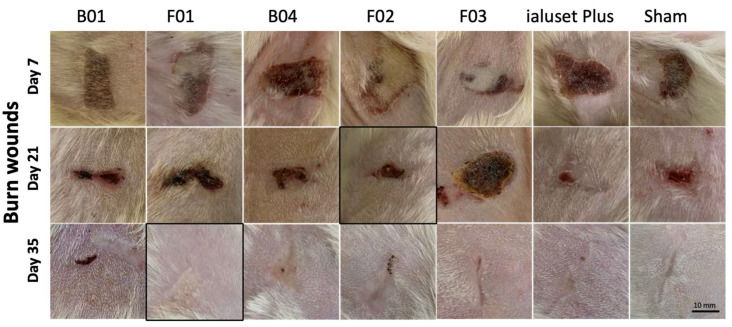
Pictures of burn wound healing in Sprague Dawley rats receiving DES formulations. Wounds of study groups observed at weeks 1 (day 7), 3 (day 21) and 5 (day 35). Unhealed wounds at week 1 with significant retraction of burn area in all groups, except F03, and scar formation at week 5. F01 showed less scarring in burn wounds than any other group, including the positive control. Photographs in boxes are those most clinically healed at time of observation in comparison to the positive control.

**Figure 9 molecules-28-02402-f009:**
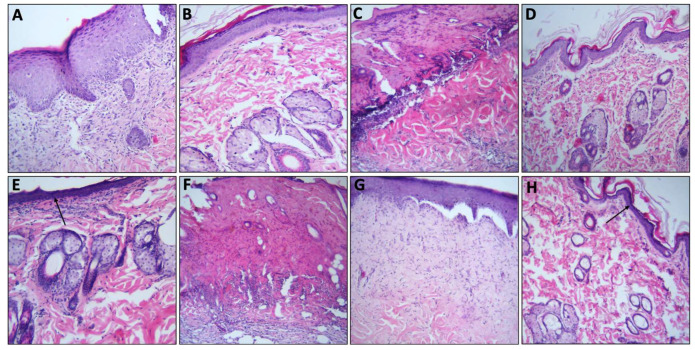
Histological evaluation of burn wounds at different periods. (**A**) Wound from sham group at 7 days, showing epithelial migration and presence of inflammatory infiltrates (neutrophils, lymphocytes, and histiocytes) (HE, ×10); (**B**) wound from positive control “ialuset” at 28 days showing full re-epithelialisation process and normal distribution of infiltrates; (**C**) wound from F01 at 7 days with the presence of a serohematic crust, blood vessels, and presence of inflammatory infiltrates (neutrophils, lymphocytes, and histiocytes) (HE, ×10); (**D**) wound from F01 with complete re-epithelialisation process, new blood vessels and hair follicles at 28 days showing reduced inflammatory infiltrate (HE, ×10); (**E**) F03 at 28 days showing reducing inflammatory infiltrates with new blood vessels and hair follicles, with mild scarring presented by a darkened epidermis (arrow); (**F**) wound from F02 at 7 days with the presence of a serohematic crust, presence of inflammatory infiltrates (neutrophils, lymphocytes, and histiocytes) (HE, ×10); (**G**) wound in re-epithelialisation process from F02 treated group at 28 days with reduced presence of inflammatory infiltrate, and absence of complete re-epithelialisation and thick epidermis (HE ×10); (**H**) wound in re-epithelialisation process from F02 group after 28 days showing reducing inflammatory infiltrates with new blood vessels and mild scarring presented by a darkened epidermis (arrow) (HE ×10).

**Table 1 molecules-28-02402-t001:** Characterisation of blank DES formulations. MA: Malonic acid, CC: Choline chloride.

Entry	Composition	Ratio	Spreadability Cm (mean ±SD)	Contact Angle (θ)
B01	MA:CC	1:1	3.51 ± 0.13	NM **
B02	MA:CC	1:2	2.51 ± 0.13	NM **
B03	MA: CC	2:1	1.61 ± 0.13	NM **
B04	B01:PG	1:1	6.0 ± 0.12	70 ± 2.9
B05	B01:PG	1:2	7.2 ± 0.11	67 ± 2.4
B06	B02:PG	1:1	4.1 ± 0.13	79 ± 3.4
B07	B02:PG	1:2	4.9 ± 0.16	76 ± 2.9
ialuset Plus	NA *	NA *	6.3 ± 0.11	NM **

* NA: not applicable, ** NM: not measured. ialuset Plus: a commercial product for wound healing.

**Table 2 molecules-28-02402-t002:** The detailed composition of the prepared formulations of TDF and LDC.

DES Formulation	Composition	Molar Ratio of Drugs	Vehicle
**F01**	TDF and LDC	1:3	B01
**F02**	TDF and LDC	1:3	B04
**F03**	TDF	-	B04

**Table 3 molecules-28-02402-t003:** MIC values (µL/mL) of B01 and F01 against a panel of microorganisms.

	MIC in Liquid Broth (µL/mL)	
Strain	B01	F01
*Enterococcus faecalis* ATCC19422	5	5
*Staphylococcus aureus* NCTC 6571	5	5
*Escherichia coli* MC4100	5	5
*Pseudomonas aeruginosa* PAO1	2.5	5
*Candida albicans* SC5314	>10*	>10 *

* >10 µL/mL: no inhibition at the highest concentration that was tested.

## Data Availability

Not applicable.

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
