# Peer review of "Deep Eutectic Liquids as a Topical Vehicle for Tadalafil: Characterisation and Potential Wound Healing and Antimicrobial Activity"

_molecules, 2023, doi:10.3390/molecules28052402_

Round 1

Reviewer 1 Report

The paper is well-written and organized and it contains interesting results.

1. I would like to suggest that the authors include an abbreviation list and the chemical structures for the drugs used in the study.

2. Check the references to figures from text. For example:

-line 238 contains a reference to figure 3C as containing a DSC curve but figure 3C from the manuscript contains an NMR spectrum;

- lines 434-435 make reference to NMR spectra from figures 1 and 2 but these figures contain rheology and solubility results.

Author Response

Please see the attachment for all reviewers comments

Reviewer 2 Report

Review of molecules-2214506 entitled “Deep Eutectic liquids as a Topical Vehicle of Tadalafil: Characterization and Potential Wound Healing and Antimicrobial Activity”

You need to define F01, F02, B0X exhaustively before the first appearance, not like below:

r.29 – “… forming F01. The addition of propylene glycol (PG) to the formulation was attempted to reduce the viscosity, referred to as F02.”

r.117 – “…B03 ˃B01 ˃ B02.

Be consistent:

““ialuset Plus” and (ialuset Plus) vs Ialuset Plus

(table 1) vs (Figure 1)

Figure 1 – use logarithmic scale for Y axis.

Figure 3,5 – The text and digits on the figures are practically invisible

Figure 4 – Some digits on the figure is practically invisible

r.388 “was circulated through C-18 column with particle size of 5 μm (150 mm x 4.6 mm).” Could you clarify it?

Conclusion: resubmit after revision

Author Response

(The authors gave the same response as above.)

Reviewer 3 Report

In this study, the authors designed a CC-based delivery system (DESs) of tadalafil (TDF), which is a selective phosphodiesterase type 5 (PDE-5) enzyme inhibitor, for wound healing purposes. The results demonstrated the utility of the F01 in wound healing both in vitro and in vivo using cut wound and burn wound models.

To further improve the study, it is recommended that the authors

1. Please add scale bars in figures 6, 8, and 9.

2. Additionally, before conducting the in vivo animal study, it would be beneficial to perform an in vitro study to evaluate the cell toxicity of the materials.

Author Response

(The authors gave the same response as above.)

Reviewer 4 Report

The present manuscript proposes deep eutectic solvents (DESs) formed by choline chloride (CC) and malonic acid as vehicles for a topical formulation of tadalafil.

Paragraph 3.4.1 it was not reported how viscosity values has been calculated from flow curve. Which geometry was used for the viscosity measurements?

Line 410 What are OCA and PCA?

Paragraph 3.11 It is not clear how the formulation was applied on rat skin? Which is the applied dose? How many times the formulation was applied on rat skin during the six weeks of treatment?

The rationale about the inclusion in the formulation of lidocaine should be better explained as the effect of lidocaine in increasing tadalafil solubility in DESs.

Cytological and tissue toxicity  of the prepared DESs should be addressed

Author Response

(The authors gave the same response as above.)

Round 2

Reviewer 4 Report

Authors reply: The reported values of viscosity in the manuscript was obtained from the raw data obtained from the rheometer. For comparison purposes a temperature of 25 C was fixed for the formulations.

Reviewer: It should be reported how the viscosity values were calculated from the raw data. For instance, by fitting the raw curve with a model as “power law” or by doing the mean of viscosity values at different shear rates or by reporting the viscosity at a selected shear rate value for each flow curve.

Author reply: A thin layer of formulations was applied aseptically once daily for 6 consecutive weeks. A moderate quantity was applied and distributed over the wound using sterile gloved hands to mimic the clinical conditions.

Reviewer. What is a “moderate quantity”? An exact dose must be applied on rats and reported. Otherwise, the study is not reproducible.

Author Response

1. 

  1. Reviewer: It should be reported how the viscosity values were calculated from the raw data. For instance, by fitting the raw curve with a model as “power law” or by doing the mean of viscosity values at different shear rates or by reporting the viscosity at a selected shear rate value for each flow curve.

        Response: The viscosity was reported at a selected shear rate value for each flow curve.

The following statement was added to the text as appeared as red track changes in the text of the manuscript. 

"at a shear rate value of 50 s-1"

2. Reviewer comment:  What is a “moderate quantity”? An exact dose must be applied to rats and reported. Otherwise, the study is not reproducible.

Response: Exactly 1g quantity of tested formulation and reference product was applied.

The above statement was added to the text of the manuscript and appeared as red track changes.